# PREDICATED DIFFUSION: PREDICATE LOGIC-BASED ATTENTION GUIDANCE FOR TEXT-TO-IMAGE DIFFUSION MODELS

## ABSTRACT

Diffusion models have achieved remarkable results in generating high-quality, diverse, and creative images. However, when it comes to text-based image generation, they often fail to capture the intended meaning presented in the text. For instance, a specified object may not be generated, an unnecessary object may be generated, and an adjective may alter objects it was not intended to modify. Moreover, we found that relationships indicating possession between objects are often overlooked. While users' intentions in text are diverse, existing methods tend to specialize in only some aspects of these. In this paper, we propose Predicated Diffusion, a unified framework to express users' intentions. We consider that the root of the above issues lies in the text encoder, which often focuses only on individual words and neglects the logical relationships between them. The proposed method does not solely rely on the text encoder, but instead, represents the intended meaning in the text as propositions using predicate logic and treats the pixels in the attention maps as the fuzzy predicates. This enables us to obtain a differentiable loss function that makes the image fulfill the proposition by minimizing it. When compared to several existing methods, we demonstrated that Predicated Diffusion can generate images that are more faithful to various text prompts, as verified by human evaluators and pretrained image-text models.

## 1 INTRODUCTION

The recent advancements in deep learning have paved the way for the generation of images that are high-quality, diverse, and creative. This progress is primarily attributed to diffusion models (Ho et al., 2020; Song et al., 2021), which recursively update images to remove noise and to make them more realistic. Diffusion models are significantly more stable and scalable compared to previous methods, such as generative adversarial networks (Goodfellow et al., 2014; Radford et al., 2016) or autoregressive models (van den Oord et al., 2016; Kolesnikov & Lampert, 2017). Moreover, the field of text-based image generation is attracting considerable attention, with the goal being to generate images that are faithful to a text prompt given as input. Even in this area, the contributions of diffusion models are notable (Ramesh et al., 2021). We can benefit from commercial applications such as DALL-E2 (Ramesh et al., 2022) and Imagen (Saharia et al., 2022), as well as the state-of-the-art open-source model, Stable Diffusion (Rombach et al., 2022). These models are trained on large-scale and diverse text-image datasets, which allows them to respond to a variety of prompts and to generate images of objects with colors, shapes, and materials not found in the existing datasets.

However, many previous studies have pointed out that these models often generate images that ignore the intended meanings of a given prompt, as exemplified in Fig. 1 (Feng et al., 2023; Chefer et al., 2023; Rassin et al., 2023; Wang et al., 2023). When multiple objects are specified in a prompt, only some are generated, with the others disappearing (see the column *missing objects* in Fig. 1). Also, two specified objects are sometimes mixed together to form one object in the generated image (*object mixture*). Given an adjective in a prompt, it alters a different object than the one the adjective was originally intended to modify (*attribute leakage*). We have found a novel challenge that, when a prompt specifies an object being held by someone, the object is depicted as if discarded on the ground (*possession failure*). Although these challenges need to be addressed, retraining diffusion models on large-scale datasets is prohibitively expensive. Many studies have proposed methods

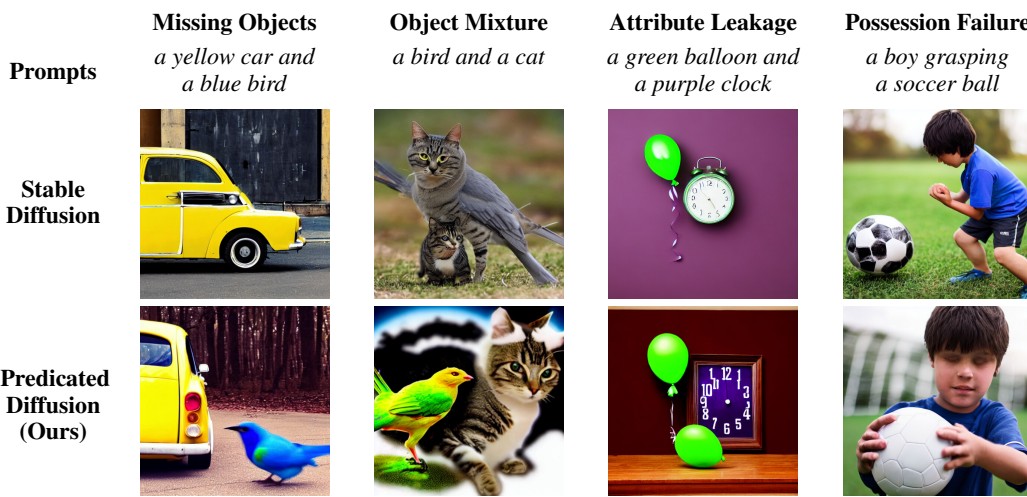

Figure 1: Visualizations of typical challenges in text-based image generation using diffusion models. The proposed Predicated Diffusion can solve all of these challenges, but is not limited to them.

offering guidance for the image generation process of pre-trained diffusion models, ensuring that the images are updated to become more faithful to the prompt. However, these guidances vary widely, and a unified solution to address the diverse range of challenges has yet to be established.

We hypothesize that the root cause of such challenges lies in the text encoder within diffusion models failing to correctly capture the logical statements presented in the given prompt. If we could represent such logical statements by using predicate logic and integrate it into the diffusion model, it might make the generated images more faithful to the statements. Motivated by this idea, we introduce *Predicated Diffusion* in this paper. Herein, we represent the relationships between the words in the prompt by propositions using predicate logic. By employing attention maps and fuzzy logic (Hájek, 1998; Prokopowicz et al., 2017), we measure the degree to which the image under generation fulfills the propositions, providing guidance for images to become more faithful to the prompt. See the conceptual diagram in Fig. 2. The contribution of this paper is threefold.

**Theoretical Justification and Generality:** Most existing methods have been formulated based on deep insights, which makes it unclear how to combine them effectively or how to apply them in slightly different situations. In contrast, Predicated Diffusion can resolve a variety of challenges based on the same foundational theory, allowing us to deductively expand it to address challenges not summarized in Fig. 1.

**High Fidelity to Prompt:** The images generated by the proposed Predicated Diffusion and comparison method were examined by human evaluators and pretrained image-text models (Radford et al., 2021; Li et al., 2022). We confirmed that Predicated Diffusion generates images that are more faithful to the prompts and is more likely prevent the issues shown in Fig. 1.

**New Challenge and Solution:** This paper introduces a new challenge, named *possession failure*, which occurs when the generated image fails to correctly depict a prompt indicating a subject in possession of an object. Thus, we broaden the horizons of the current research, which has mainly focused on the presence or absence of objects and attributes, to encompass actions. The fact that Predicated Diffusion can successfully address this new challenge is worthy of attention.

## 2 RELATED WORK

**Conditional Image Generation**   A diffusion model was proposed as a parameterized Markov chain (Sohl-Dickstein et al., 2015; Ho et al., 2020). Taking a given image $x$ as the initial state $x_0$, the forward process $q(x_{t+1}|x_t)$ adds noise to the state $x_t$ repeatedly. The model learns the reverse process $p(x_{t-1}|x_t)$, reproducing the data distribution $p(x) = p(x_0)$. Intuitively speaking, it repeatedly denoises images to be more realistic. The reverse process resembles a discretized

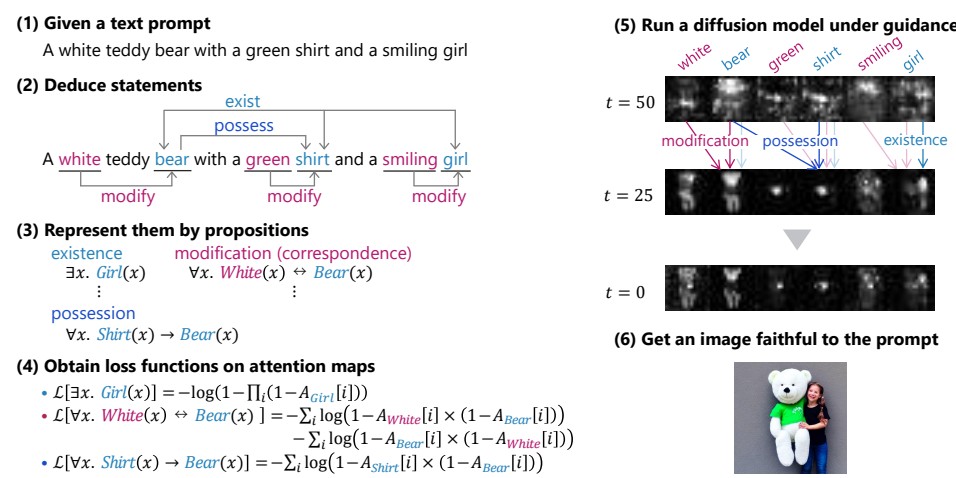

Figure 2: The conceptual diagram of the proposed Predicated Diffusion, composed of steps (1)–(6).

stochastic differential equation, akin to the Langevin dynamics, which ascends the gradient of the log-probability, $\nabla \log p(x)$ (Song et al., 2021). With a separate classifier $p(y|x)$ for class label $y$, the diffusion model can reproduce the conditional probability $p(x|y)$, by ascending the gradient of the conditional log-probability, $\nabla \log p(x|y) = \nabla \log p(y|x) + \nabla \log p(x)$. Although grounded in probability theory, what it practically offers is additional guidance $\nabla \log p(y|x)$ for updating images, which is generalized as *classifier guidance* (Dhariwal & Nichol, 2021). The diffusion model can learn the conditional probability $p(x|c)$ directly. The condition $c$ might be text, images, or other annotations (Ramesh et al., 2021; Rombach et al., 2022). The difference between conditional and unconditional updates serves as *classifier-free guidance*, which can adjust the fidelity of the generated image to condition $c$ (Ho & Salimans, 2021). Liu et al. (2022) proposed Composable Diffusion, inspired by energy-based models (Du et al., 2020). It generates an image conditioned on two concepts, $c_0$ and $c_1$, by summing their respective conditional updates. It negates or removes a concept $c_n$ from generated images by subtracting the update conditioned on $c_n$, termed as a *negative prompt*.

**Text-Based Image Generation by Cross-Attention Mechanism**  One of the leading models, Stable Diffusion, employs the cross-attention mechanism for conditioning (Vaswani et al., 2017). A convolutional neural network (CNN), U-Net (Ronneberger et al., 2015), transforms the image $x$ into an intermediate representation. For text conditions, a text encoder, CLIP (Radford et al., 2021), transforms text prompt $y$ into a sequence of intermediate representations, each linked to a word $c$ within the prompt $y$. Given these representations, the cross-attention mechanism creates an attention map $A_c$ for each word $c$. Using these maps as weights, U-Net then updates the image $x$. Technically, these processes target not the image $x$ but the latent variable $z$ extracted by a variational autoencoder (Kingma & Welling, 2014). Despite its sophistication, Stable Diffusion sometimes fails to capture the intended meaning of the text prompt, as discussed in the Introduction.

The novelty of Stable Diffusion primarily lies in its structure, which is compatible with existing guidances such as Composable Diffusion and has inspired new guidances. Structure Diffusion feeds segmented text prompts to the text encoder to emphasize each clause (Feng et al., 2023). High pixel intensity in the attention map $A_c$ suggests the presence of a corresponding object or concept $c$ at that pixel. Hence, recent studies offer guidances based on the attention map, termed as *attention guidance*. Attend-and-Excite enhances the intensity of at least one pixel in the attention map $A_c$ to ensure the existence of the corresponding object $c$ (that is, address missing objects) (Chefer et al., 2023). SynGen equalizes the intensity distributions for related nouns and adjectives, while differentiating others, thus addressing attribute leakage (Rassin et al., 2023). While these methods are based on deep insights, they lack comprehensive theoretical justification and generality.

Another branch of studies has proposed attention guidances using external annotations, such as bounding boxes (Xie et al., 2023; Ma et al., 2023; Mao & Wang, 2023) and segmentation masks (Park et al., 2023). While effective in intentionally controlling image layout, these methods sometimes limit the diversity of the generated images.

Table 1: Propositions and Attention Map.

| Proposition | Attention Map |
|---|---|
| true | $1$ |
| false | $0$ |
| $P(x)$ | $A_P[i]$ |
| $\neg P(x)$ | $1 - A_P[i]$ |
| $P(x) \wedge Q(x)$ | $A_P[i] \times A_Q[i]$ |
| $P(x) \vee Q(y)$ | $1 - (1 - A_P[i]) \times (1 - A_Q[j])$ |
| $P(x) \rightarrow Q(x)$ | $1 - A_P[i] \times (1 - A_Q[i])$ |
| $\forall x.\, P(x)$ | $\prod_i A_P[i]$ |
| $\exists x.\, P(x)$ | $1 - \prod_i (1 - A_P[i])$ |

Table 2: Statements that Predicated Diffusion Can Express.

| Statements | Example Prompts | Loss |
|---|---|---|
| Existence | There is a dog | (1) |
| Concurrent existence | There are a dog and a cat | (2) |
| Adjective | A black dog | (3) |
| One-to-one correspondence | A black dog and a white cat | (5) |
| Possession | A man holding a bag | (6) |
| Multi-color | A green and grey bird | (A1) |
| Negation | without snow | (A2) |

## 3 METHOD: PREDICATED DIFFUSION

**Predicate Logic** First-order predicate logic is a formal language for expressing knowledge (Genesereth & Nilsson, 1987). Variables like $x$ and $y$ denote unspecified objects. Predicates like $P$ and $Q$ indicate properties or relationships between objects. Using variables and predicates, we can express logical statements that define object properties. For example, the proposition $P(x)$ represents the statement that "$x$ has property $P$." If the predicate $P$ indicates the property "being a dog," the proposition $P(x)$ represents the statement that "$x$ is a dog." The existential quantifier, denoted by $\exists$, declares the existence of objects satisfying a given property. Thus, the proposition $\exists x.\, P(x)$ asserts the existence of at least one object $x$ that satisfies the predicate $P$, representing that "There is a dog."

**Predicate Logic in Attention Map and Resulting Gauidance** Through predicate logic, we propose an attention-based guidance termed *Predicated Diffusion*. In a diffusion model for text-based image generation, a cross-attention mechanism creates attention maps. Each map is linked to a word in a text prompt and assigns weights to specific regions of the image. We denote the attention map linked to the word $P$ by $A_P$. The value $A_P[i] \in [0, 1]$ refers to the intensity of the $i$-th pixel in $A_P$. We treat the intensity $A_P[i]$ as a continuous form of a proposition $P(x)$. $A_P[i] = 1$ indicates that the proposition $P(x)$ holds, whereas $A_P[i] = 0$ implies that it does not. $1 - A_P[i]$ indicates the the negation of the proposition, $\neg P(x)$. The correspondences are summarized in Table 1, which are inspired by the strong conjugation, strong negation, and material implication of the product fuzzy logic (Hájek, 1998; Prokopowicz et al., 2017).

Given another proposition $Q(x)$, we consider the conjunction $Q(x) \wedge P(x)$ to correspond to the product $A_Q[i] \times A_P[i]$ in the attention maps. One can derive any logical operations using both negation and conjunction. For example, because the disjunction $Q(x) \vee P(x)$ is equivalent to $\neg(\neg Q(x) \wedge \neg P(x))$, it corresponds to $1 - (1 - A_Q[i]) \times (1 - A_P[i])$. The universal quantifier $\forall$ asserts that a predicate holds for all object. Thus , $\forall x.\, P(x) = \wedge_x P(x)$ corresponds to $\prod_i A_P[i]$. Using this, the existential proposition $\exists x.\, P(x)$ can be re-expressed as $\neg(\forall x.\, \neg P(x))$, corresponding to $1 - \prod_i (1 - A_P[i])$.

For simplicity, we will treat italicized nouns and adjectives as predicates. Specifically, we will use $Dog(x)$ to represent "$x$ is a dog" rather than $P(x)$. A text prompt "There is a dog" is represented by the proposition $\exists x.\, Dog(x)$. Then, we expect that $1 - \prod_i (1 - A_{Dog}[i]) = 1$. To encourage this, we consider its negative logarithm,

$$\mathcal{L}[\exists x.\, Dog(x)] = -\log(1 - \prod_i (1 - A_{Dog}[i])), \qquad (1)$$

and adopt it as the loss function, making the intensity of at least one pixel approach 1. This loss function is inspired by the negative log-likelihood for Bernoulli random variables. Moving forward, we will denote the loss function resulting from the proposition $R$ by $\mathcal{L}[R]$. We provide an overview of prompts and their corresponding loss functions in Table 2.

The reverse process of a diffusion model, $q(x_{t-1}|x_t, c)$, is typically modeled as a Gaussian distribution $q(x_{t-1}|x_t, c) = \mathcal{N}(x_{t-1}|\mu_\theta(x_t, t, c), \Sigma_\theta(x_t, t, c))$. The parameters for this are determined by neural networks $\mu_\theta$ and $\Sigma_\theta$ which consider the current image $x_t$, time $t$, and condition $c$. The attention map $A_P$ serves as a component of these neural networks. We take the gradient of the loss function with respect to the input image, $\nabla_{x_t} \mathcal{L}[R]$, and subtract it from the mean of the reserve process as $q(x_{t-1}|x_t, c) = \mathcal{N}(x_{t-1}|\mu_\theta(x_t, t, c) - \nabla_{x_t} \mathcal{L}[R], \Sigma_\theta(x_t, t, c))$, which decreases the loss

function $\mathcal{L}[R]$ and guides the image toward fulfilling the proposition $R$. This modification of the reverse process is referred to as guidance. In general, to encourage a proposition to hold, one can convert it to an equation of the attention map intensity, take its negative logarithm, use it as a loss function, and integrate it into the reverse process. A visual representation of this is found in Fig. 2.

**Concurrent Existence by Logical Conjunction** In practice, when text prompts include multiple objects, one of the objects often disappears. Take, for instance, the prompt "There are a dog and a cat." This can be decomposed into two statements: "There is a dog" and "There is a cat." Given that a set of statements can be represented through the conjunction of propositions, the prompt can be represented by the proposition $(\exists x. Dog(x)) \wedge (\exists x. Cat(x))$. The corresponding loss function is

$$\mathcal{L}[(\exists x. Dog(x)) \wedge (\exists x. Cat(x))] = \mathcal{L}[\exists x. Dog(x)] + \mathcal{L}[\exists x. Cat(x)]. \tag{2}$$

Minimizing this loss function encourages the concurrent existence of both a dog and a cat.

**Adjective by Logical Implication** We develop these ideas into logical implication. For a prompt such as "There is a black dog," it can be decomposed into: "There is a dog" and "The dog is black." The former statement has been previously discussed. The latter can be represented with the proposition $\forall x. Dog(x) \rightarrow Black(x) = \forall x. \neg(Dog(x) \wedge \neg Black(x))$. Thus, the loss function is

$$\mathcal{L}[\forall x. Dog(x) \rightarrow Black(x)] = -\sum_i \log(1 - A_{Dog}[i] \times (1 - A_{Black}[i])). \tag{3}$$

To ensure the both statements, we can sum the loss functions (1) and (3).

**One-to-One Correspondence** As far as we have confirmed, the existing models rarely fail to generate an object with a specified color. These models might struggle when handling prompts with multiple adjectives and nouns; one of the specified objects may not be generated properly, one of the specified adjectives may be ignored, or an adjective may modify a wrong noun. The first two issues can be addressed using the loss functions (2) and (3). The last issue is often referred to as attribute leakage. For example, given the prompt "a black dog and a white cat," leakage could lead to the generation of a white dog or a black cat. To prevent leakage, we must deduce statements implicitly suggested by the original prompt. From the prompt, we can deduce not only "The dog is black" but also "The black object is a dog." The latter can be represented by the proposition $\forall x. Black(x) \rightarrow Dog(x)$. When combined, these two statements can be represented as a biimplication: $\forall x. Dog(x) \leftrightarrow Black(x) = (\forall x. Dog(x) \rightarrow Black(x)) \wedge (\forall x. Black(x) \rightarrow Dog(x))$. This leads to the loss function

$$\mathcal{L}[\forall x. Dog(x) \leftrightarrow Black(x)] = \mathcal{L}[\forall x. Dog(x) \rightarrow Black(x)] + \mathcal{L}[\forall x. Black(x) \rightarrow Dog(x)]. \tag{4}$$

Furthermore, we can deduce a negative statement, "The dog is not white," represented by $\forall x. Dog(x) \rightarrow \neg White(x)$. Thus, the comprehensive loss function for the original statement is:

$$\begin{aligned}\mathcal{L}_{\text{one-to-one}} = &\mathcal{L}[\forall x. Dog(x) \leftrightarrow Black(x)] + \mathcal{L}[\forall x. Cat(x) \leftrightarrow White(x)] \\ &+ \alpha\mathcal{L}[\forall x. Dog(x) \rightarrow \neg White(x)] + \alpha\mathcal{L}[\forall x. Cat(x) \rightarrow \neg Black(x)],\end{aligned} \tag{5}$$

where the hyperparameter $\alpha \in [0, 1]$ adjusts the weight of the implicitly negative statements. To further ensure the existence of objects, the loss function (2) can also be applied.

**Possession by Logical Implication** We introduce another type of implication. Consider the text prompt "a man holding a bag." This implies that the *bag* forms part of the *man*. Such a relationship can be represented by the proposition $Bag(x) \rightarrow Man(x)$, leading to the loss function

$$\mathcal{L}[\forall x. Bag(x) \rightarrow Man(x)] = -\sum_i \log(1 - A_{Bag}[i] \times (1 - A_{Man}[i])). \tag{6}$$

Not limited to the word *holding*, other words indicating possession such as *having*, *grasping*, and *wearing* can be represented by the logical implication.

**Discussions and Potential Extensions** Several studies have introduced loss functions or quality measures for machine learning methods by drawing inspiration from fuzzy logic (Hu et al., 2016; Diligenti et al., 2017; Mordido et al., 2021; Marra et al., 2023) (see also Giunchiglia et al. (2022) for a survey). In this context, Predicated Diffusion is the first method to establish the correspondence between the attention map and the predicates. The propositions and corresponding loss functions

can be adapted to a variety of scenarios, including, but not limited to, the concurrent existence of more than two objects, a single object modified by multiple adjectives, the combination of one-to-one correspondence and possession, and the negation of existence, modifications, and possessions, as we will show in the following sections. Some previous studies extracted the structure of sentences using syntactic parsers or obtained relationships between words from additional data such as scene graphs (Feng et al., 2023). Such methods can be combined with Predicated Diffusion.

The (weak) conjugation of Gödel fuzzy logic and the product fuzzy logic is achieved by the minimum operation (Hájek, 1998; Prokopowicz et al., 2017). If we employ this operation and define the loss function by taking the negative instead of the negative logarithm, the proposition asserting the concurrent existence, $(\exists x.\, Dog(x)) \wedge (\exists x.\, Cat(x))$, leads to the loss function $\max(1 - \max_i A_{Dog}[i], 1 - \max_i A_{Cat}[i])$. This is equivalent to the one used for Attend-and-Excite (Chefer et al., 2023). This comparison suggests that our approach considers Attend-and-Excite as Gödel fuzzy logic, replaces the underlying logic with the product fuzzy logic, and broadens the scope of target propositions. Similar to the loss function (5), SynGen equalizes the attention map intensities for related nouns and adjectives (Rassin et al., 2023). SynGen additionally differentiates those for all word pairs except for the adjective-noun pairs. In contrast, the loss function (5) differentiates those for only specific pairs which could trigger attribute leakage based on inferred propositions, thereby preventing the disruption of the harmony, as shown in the following section.

## 4 Experiments and Results

**Experimental Setting**  We implemented Predicated Diffusion by adapting the official implementation of Attend-and-Excite (Chefer et al., 2023)[1]. The reverse process spans 50 steps; following Attend-and-Excite and SynGen, we applied the guidance of Predicated Diffusion only to the initial 25 steps. See Appendix A.1 for more details. For comparative evaluation, we also prepared Composable Diffusion (Liu et al., 2022), Structure Diffusion (Feng et al., 2023), and SynGen (Rassin et al., 2023), in addition to Stable Diffusion and Attend-and-Excite. All models used the officially pretrained Stable Diffusion (Rombach et al., 2022)[2] as backbones.

We conducted four experiments for assessing each method's performance. We provided each method with the same prompt and random seed, and then generated a set of images. Human evaluators were tasked with the visual assessment of these generated images. Instructions and evaluation criteria provided to the evaluators are detailed in Appendix A.2.

(i) *Concurrent Existence*: We prepared 400 random prompts, each mentioning "[Object A] and [Object B]", and generated 400 sets of images. The evaluators identified cases of "missing objects," where one or both of the specified two objects were not generated. Some cases involved an "object mixture", where, although the two objects were generated, their boundaries were unclear. The evaluators tallied the cases of "missing objects" based on two criteria: a lenient criterion where "object mixture" was not counted as "missing objects", and a strict criterion where it was. For Predicated Diffusion, we used the loss function (2).

(ii) *One-to-One Correspondence*: Similarly, we prepared 400 random prompts, each mentioning "[Adjective A] [Object A] and [Adjective B] [Object B]". In addition to identifying missing objects, the evaluators identified the cases where adjectives incorrectly altered unrelated objects and tallied the number of such cases as "attribute leakage". For Predicated Diffusion, we used the loss function (2)+(5) with $\alpha = 0.3$.

(iii) *Possession*: We prepared 10 prompts, each mentioning "[Subject A] is [Verb C]-ing [Object B]". [Verb C] can be "have," "hold," "wear," or the like. We generated 20 images for each of these prompts. In addition to identifying missing objects, the evaluators identified the cases where [Verb C] was not executed appropriately, and tallied the number of such cases as "possession failure". For Predicated Diffusion, we used the loss function (2)+(6).

(iv) *Complicated*: To demonstrate the generality of Predicated Diffusion, we prepared diverse prompts, some of which were taken from the ABC-6K dataset (Feng et al., 2023). Images were generated after manually extracting propositions and their respective loss functions. While a summary of generated images is presented, numerical evaluations were not undertaken due to the diversity of the prompts.

---

[1] https://github.com/yuval-alaluf/Attend-and-Excite (MIT license)
[2] https://github.com/CompVis/stable-diffusion (CreativeML Open RAIL-M)

Table 3: Results of Experiments (i) and (ii)

| Models | Experiment (i) | | | Experiment (ii) | | | |
|---|---|---|---|---|---|---|---|
| | Missing[†] Objects | Fidelity | Similarity[‡] | Missing[†] Objects | Attribute Leakage | Fidelity | Similarity[‡] |
| Stable Diffusion | 54.7 / 66.0 | 11.0 | 0.325 / 0.770 | 64.8 / 73.5 | 88.5 | 6.0 | 0.343 / 0.741 |
| Composable Diffusion | 44.5 / 82.3 | 2.5 | 0.318 / 0.740 | 49.3 / 83.5 | 88.5 | 3.8 | 0.347 / 0.725 |
| Structure Diffusion | 56.0 / 64.5 | 12.0 | 0.320 / 0.760 | 64.3 / 69.5 | 86.5 | 5.8 | 0.342 / 0.737 |
| Attend-and-Excite | 25.3 / 36.3 | 29.5 | 0.337 / 0.814 | 28.0 / 35.8 | 64.5 | 19.3 | 0.367 / 0.781 |
| SynGen | — | — | — | 23.3 / 29.3 | 40.3 | 36.8 | 0.365 / 0.792 |
| Predicated Diffusion | **18.5 / 28.5** | **30.3** | **0.340 / 0.837** | **10.0 / 16.5** | **33.0** | **44.8** | **0.375 / 0.808** |

[†]Using the lenient and strict criterions. [‡]Text-image similarity and text-text similarity.

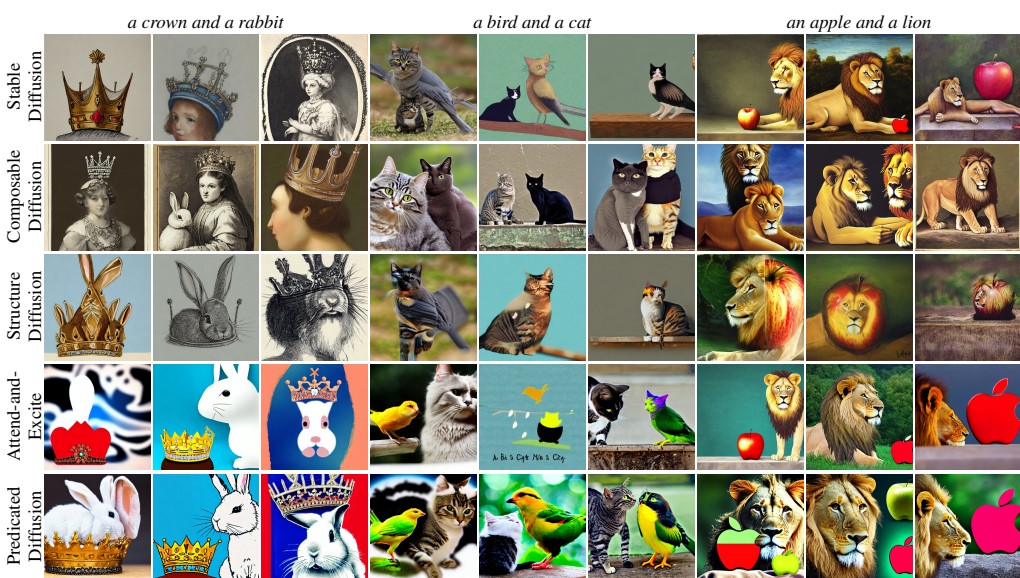

Figure 3: Example results of Experiment (i) for concurrent existence. See also Fig. A2.

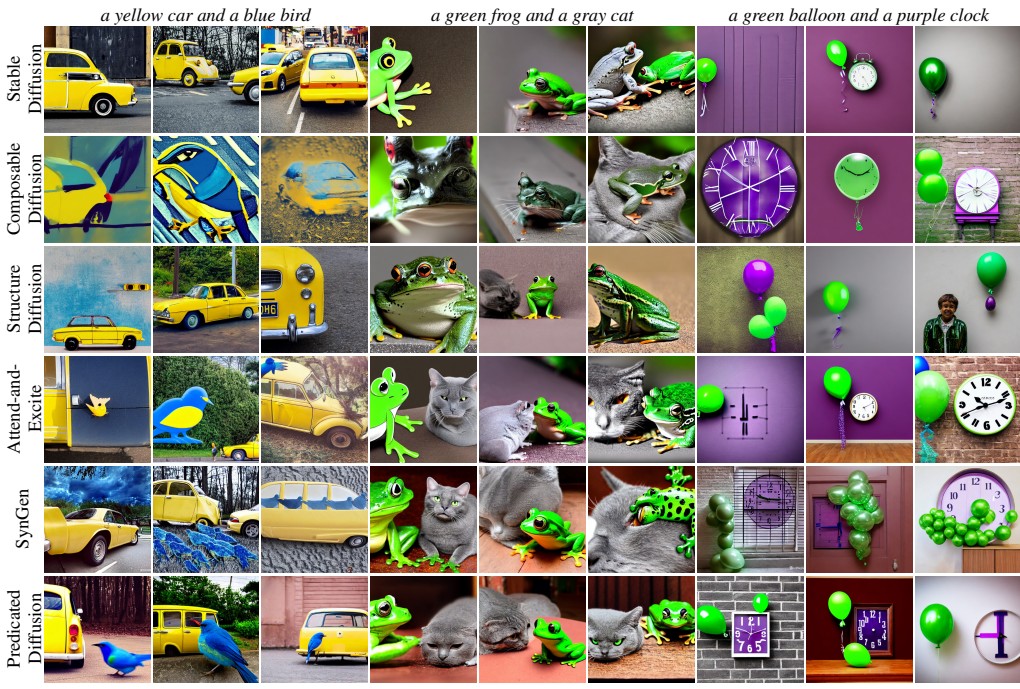

Figure 4: Example results of Experiment (ii) for one-to-one correspondence. See also Fig. A3.

Table 4: Results of Experiment (iii) for possession

| Models | Missing Objects[†] | Possession Failure | Fidelity |
|---|---|---|---|
| Stable Diffusion | 31.5 / 36.0 | 52.5 | 33.5 |
| Attend-and-Excite | 7.5 / 17.0 | 51.5 | 27.5 |
| Predicated Diffusion | **4.0 / 7.0** | **29.5** | **52.0** |

[†]Using the lenient and strict criterions.

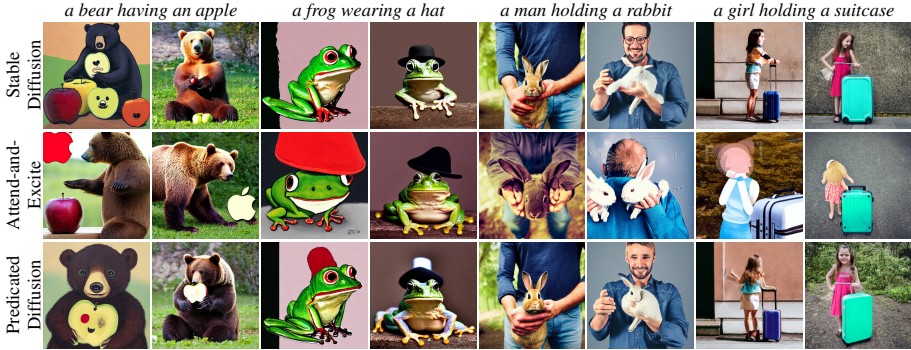

Figure 5: Example results of Experiment (iii) for possession. See also Fig. A4.

The first two experiments were inspired by previous works: Feng et al. (2023); Chefer et al. (2023); Rassin et al. (2023). In the first three experiments, the evaluators also assessed the fidelity of the generated images to the prompts. To measure fidelity automatically, we evaluated the similarity using the pretrained image-text encoder, CLIP, in two manners, following Chefer et al. (2023). We provided both the prompts and the generated images with CLIP to extract their embedding vectors, and then calculated their cosine similarity, referred to as text-image similarity. In addition, we fed the generated images into the pretrained image-captioning model, BLIP, to obtain captions (Li et al., 2022). Then, we provided both the prompts and the obtained captions with CLIP to extract their embedding vectors, and then calculated their cosine similarity, referred to as text-text similarity. The chosen objects, adjectives, and prompts can be found in Appendix A.3.

**Results for Concurrent Existence and One-to-One Correspondence** Table 3 summarizes the results of the quantitative assessments from Experiments (i) and (ii). Values except for the similarity are expressed in percentages. Higher scores are desirable for the fidelity and similarity, while lower values are preferred for the remaining criteria. Predicated Diffusion notably outperforms other methods, as it achieved the best outcomes across all 11 criteria. Figures 3 and 4 show example images for visual evaluation, where images in each column are generated using the same random seed. See also Figs. A2 and A3 in Appendix. Stable Diffusion, Composable Diffusion, and Structure Diffusion often exhibit missing objects and attribute leakage. The absent of objects is particularly evident when prompts feature unusual object combinations like "a crown and a rabbit" and "a yellow car and a blue bird." When the prompts specify visually similar objects, such as "a bird and a cat," the two objects often get mixed together. While Attend-and-Excite effectively prevents the issue of missing objects, it struggles with attribute leakage in Experiment (ii), due to its lack of a dedicated mechanism to address this. While SynGen has achieved relatively good results, Predicated Diffusion outperforms it by further preventing missing objects and attribute leakage and producing images that are the most faithful to the prompt. Although this aspect was not explicitly part of the evaluation, SynGen often generates multiple instances of small objects, such as birds and balloons.

**Results for Possession** Table 4 summarizes the results from Experiment (iii). Compared to the vanilla Stable Diffusion, Attend-and-Excite succeeds in preventing missing objects, but on the contrary, fails to prevent the possession failure and loses fidelity to the prompt. Figures 5 and A4 show visual samples of generated images. If [Subject A] is an animal, Attend-and-Excite succeeds more frequently than Stable Diffusion in depicting both objects but often depicts [Object B] as discarded on the ground or suspended in the air. If [Subject A] is a human, the vanilla Stable Diffusion often produces satisfactory results. Then, Attend-and-Excite, however, tends to deteriorate the overall

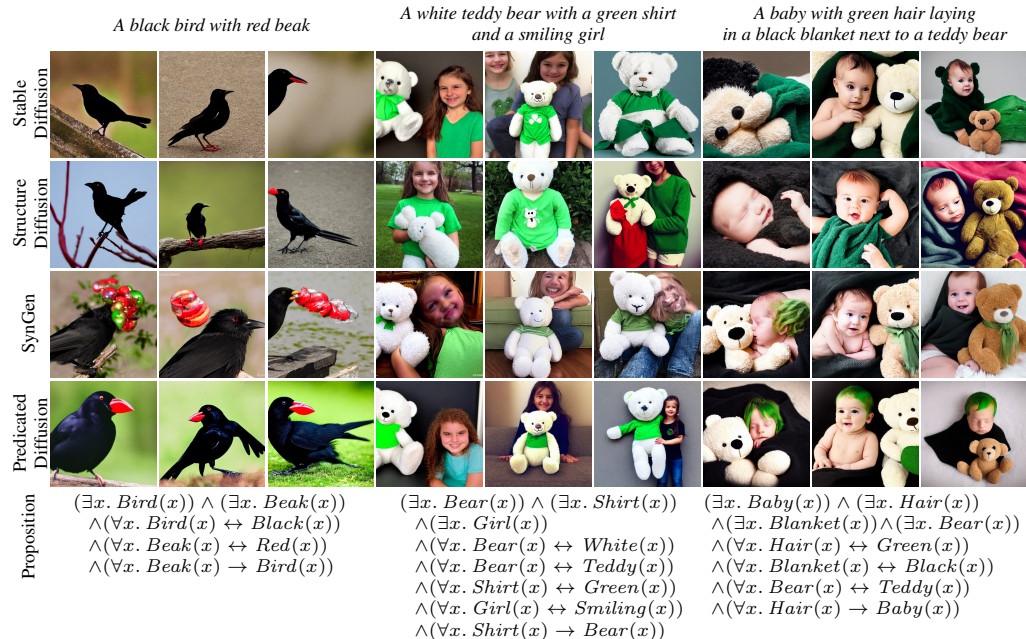

Figure 6: Example results of Experiment (iv) using prompts in ABC-6K. See also Figs. A5 and A6.

image quality. With the possession relationship, [Subject A] and [Object B] often overlap. Attend-and-Excite makes both stand out competitively and potentially disrupts the overall harmony. In contrast, the loss function (6) is designed to encourage overlap, and hence Predicated Diffusion adeptly depicts subjects in possession of objects. See Section B in Appendix for an ablation study.

**Qualitative Analysis on Complicated Prompts** Figures 6, A5, and A6 show example results from Experiment (iv) along with the propositions used for Predicated Diffusion. Both the vanilla Stable Diffusion and Structure Diffusion plagued by missing objects and attribute leakage. Experiments (i) and (ii) confirmed that SynGen often generates numerous small objects. Hence, when tasked with generating "A black bird with a red beak," it produced multiple red objects. When generating "A white teddy bear with a green shirt and a smiling girl," comparison methods other than Predicated Diffusion often mistakenly identified the girl, not the teddy bear, as the owner of the green shirt. In comparison to the vanilla Stable Diffusion, SynGen reduced the size of the teddy bear's shirt because it differentiates between the intensity distributions on the attention maps of different objects. A similar tendency is evident in the third case in Fig. 6, where the adjective "green" often modifies wrong objects, and the green hair is not placed on the baby's head. Predicated Diffusion performed well in these scenarios, which include the concurrent existence of more than two objects with specified colors and possession relationships simultaneously. See Section B in Appendix for further results and discussions, where we also examined the cases of multiple colors and negation.

## 5 CONCLUSION

This paper proposed Predicated Diffusion, where the intended meanings in a text prompt are represented by propositions using predicate logic, offering guidance for text-based image generation by diffusion models. Experiments using Stable Diffusion as a backbone demonstrated that Predicated Diffusion generates images that are more faithful to the prompt compared to other existing methods and addresses challenges observed in diffusion models: missing objects, attribute leakage, and possession failure. Moreover, due to the generality of predicate logic, Predicated Diffusion has the ability to fulfill complicated prompts that include multiple objects, adjectives, and their relationships. Although predicates cannot represent all meanings present in natural languages, they can handle most scenarios for adjusting the layout of generated images. In future work, we plan to explore the automatic extraction of propositions from prompts by syntactic parsers and explore 2-ary predicates that assert the relationships, such as $Above(x, y)$, which implies "$x$ is above $y$."

REPRODUCIBILITY STATEMENT

Details on experimental settings can be found in the first subsection of Section 4. For further information, refer to Section A in Appendix. The code and pretrained models on which our experiments rely are noted in the footnotes. We also provide the experiment code as supplementary material.

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

# A    DETAILED EXPERIMENTAL SETTING

## A.1    IMPLEMENTATION DETAILS

We confirmed that Predicated Diffusion that directly employed the loss functions in Section 3 worked well. Nonetheless, to further improve the quality of generated images, we implemented several modifications.

The loss functions are defined using either the summation or product over pixels $i$, which increase or decrease unboundedly as the image is scaled. To make these loss functions more suitable for neural networks and to prevent excessively large values, we replaced the summation with the arithmetic mean. Additionally, we replaced the finite product in (1) with the geometric mean.

Attention maps are often obtained by performing a softmax operation in the channel direction on the feature maps of a CNN, where each channel is linked to a single word or token. For calculating the loss function (1) for existence, we excluded the start-of-text token. Because this token is linked to the entire text, its omission ensures capturing the response to each individual word, consistent with the implementation of Attend-and-Excite. For calculating the loss functions (3), (5), and (6) using (bi)implication, we normalized the intensity of each attention map to a range of 0-1 using the maximum and minimum values. This allows us to focus on the relative positions, rather than the absolute intensity (that is, existence).

Drawing inspiration from Attend-and-Excite (Chefer et al., 2023), we performed the iterative refinement at $t = T$. Specifically, before executing the very first step of the reverse process, we alternated between the reverse process and the forward process four times each. Note that Attend-and-Excite performs the iterative refinement at the 10th and 20th steps until the value of the loss function falls below a certain threshold. However, we found that the very first step is crucial for modifying the layout of the generated image; therefore, we adjusted our approach as described above.

## A.2    INSTRUCTIONS TO EVALUATORS

Eight evaluators joined the experiments. They are university students aged between 19 and 21, with no background in machine learning or computer vision. The experiments were conducted in a double-blind manner: the images were presented in a random order, and the evaluators and authors were unaware of which image was generated by which model.

**Experiment (i): Concurrent Existence**    We prepared 400 random prompts, each mentioning "[Object A] and [Object B]" with indefinite articles as needed, and generated 400 sets of images.

(a) By showing one image at a time at random, we asked, "Are both specified objects generated in the image?" The evaluators answered this question with one of the following options:

1) "No object is generated."

2) "Only one of two objects is generated."

3) "Two objects are generated, but they are mixed together to form one object."

4) "Two objects are generated."

Responses 1) and 2) were categorized as "missing objects", and response 3) was categorized as "object mixture". We tallied the number of responses 1) and 2) under the lenient criterion and that of responses 1)–3) under the strict criterion.

(b) By showing a set of images generated with different models, we asked, "Which image is the most faithful to the prompt?" The evaluators were instructed to select only one image in principle, but were allowed to select more than one image if their fidelities were competitive, or not to select any image if none were faithful.

**Experiment (ii): One-to-One Correspondence**    We prepared 400 random prompts, each mentioning "[Adjective A] [Object A] and [Adjective B] [Object B]" with indefinite articles as needed.

(a) The same as above.

Table A1: Candidate Words for Generating Prompts in Experiments (i) and (ii)

| | |
|---|---|
| **Object** | cat, dog, bird, bear, lion, horse, elephant, monkey, frog, turtle, rabbit, glasses, crown, suitcase, chair, balloon, bow, car, bowl, bench, clock, apple |
| **Adjective** | red, yellow, green, blue, purple, pink, brown, gray, black, white |

Table A2: Prompts for Experiment (iii) and Results for Each Prompt

| [Subject A] | [Verb C]-ing | [Object B] | Models | *1 | *2 | *3 | *4 |
|---|---|---|---|---|---|---|---|
| rabbit | having | phone | Stable Diffusion | 12 | 12 | 15 | 5 |
| | | | Attend-and-Excite | 2 | 2 | 12 | 5 |
| | | | Predicated Diffusion | 1 | 1 | 10 | 9 |
| bear | having | apple | Stable Diffusion | 2 | 2 | 5 | 10 |
| | | | Attend-and-Excite | 0 | 1 | 10 | 1 |
| | | | Predicated Diffusion | 0 | 1 | 5 | 10 |
| monkey | having | bag | Stable Diffusion | 12 | 12 | 12 | 2 |
| | | | Attend-and-Excite | 3 | 6 | 6 | 6 |
| | | | Predicated Diffusion | 0 | 1 | 1 | 12 |
| panda | having | suitcase | Stable Diffusion | 13 | 16 | 19 | 1 |
| | | | Attend-and-Excite | 1 | 7 | 11 | 3 |
| | | | Predicated Diffusion | 1 | 2 | 8 | 7 |
| lion | wearing | crown | Stable Diffusion | 12 | 12 | 12 | 8 |
| | | | Attend-and-Excite | 0 | 0 | 0 | 20 |
| | | | Predicated Diffusion | 0 | 0 | 0 | 20 |
| frog | wearing | hat | Stable Diffusion | 9 | 10 | 10 | 3 |
| | | | Attend-and-Excite | 2 | 2 | 5 | 7 |
| | | | Predicated Diffusion | 1 | 1 | 4 | 11 |
| man | holding | rabbit | Stable Diffusion | 0 | 0 | 2 | 12 |
| | | | Attend-and-Excite | 3 | 4 | 12 | 4 |
| | | | Predicated Diffusion | 4 | 5 | 6 | 7 |
| woman | holding | dog | Stable Diffusion | 1 | 4 | 5 | 12 |
| | | | Attend-and-Excite | 3 | 8 | 16 | 2 |
| | | | Predicated Diffusion | 1 | 3 | 3 | 10 |
| boy | grasping | soccerball | Stable Diffusion | 1 | 1 | 16 | 4 |
| | | | Attend-and-Excite | 1 | 3 | 18 | 2 |
| | | | Predicated Diffusion | 0 | 0 | 14 | 6 |
| girl | holding | suitcase | Stable Diffusion | 1 | 3 | 9 | 10 |
| | | | Attend-and-Excite | 0 | 1 | 13 | 5 |
| | | | Predicated Diffusion | 0 | 0 | 8 | 12 |

*1 Missing objects in the lenient criterion, *2 Missing objects in the strict criterion,
*3 Possession failure, *4 Fidelity.

(c) At the same time as question (a), we asked, "Does each adjective exclusively modify the intended object?" The evaluators answered this question with a "Yes" or "No", and the response "No" was categorized as "attribute leakage". If one or both of the two specified objects were not generated, that is, the response to question (a) was not 4), then question (c) became irrelevant, thereby automatically marking "No" as the response.

(b) The same as above.

**Experiment (iii): Possession**   We prepared 10 prompts, each mentioning "[Subject A] is [Verb C]-ing [Object B]" with indefinite articles as needed. We generated 20 images for each of these prompts.

(a) The same as above.

(d) At the same time as question (a), we asked, "Is the [Subject A] performing [Verb C] with the [Object B]?" The evaluators answered this question with a "Yes" or "No", and the

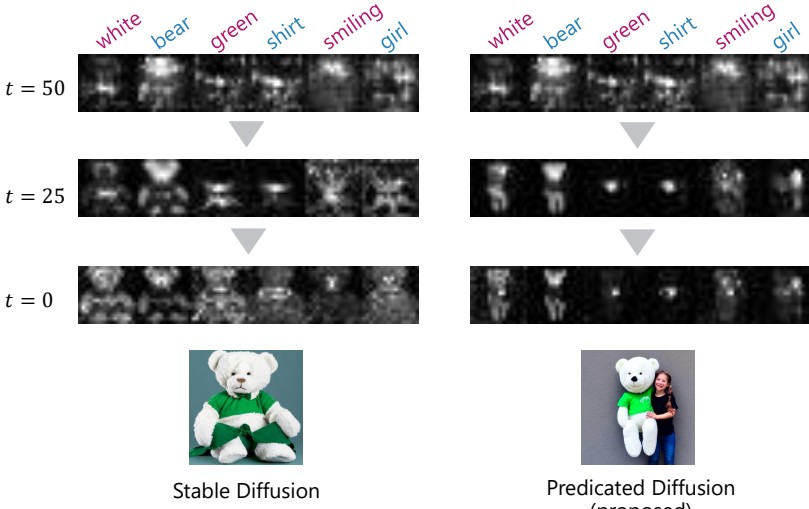

Figure A1: An example visualization of attention maps during the generation. The text prompt was "A white teddy bear with a green shirt and a smiling girl."

response "No" was categorized as "possession failure." If the response to question (a) was not 4), the response to question (d) was automatically assigned "No".

(b) The same as above.

### A.3 PROMPTS

For Experiments (i) and (ii), we randomly selected objects and adjectives from Table A1, roughly following the experiments conducted by Chefer et al. (2023). However, we excluded "mouse" and "backpack" from the list of objects, and "orange" from the list of adjectives. The term "mouse" often led to ambiguity, as it could refer to either the animal or a computer peripheral. The term "orange" also created confusion in all models, as it could indicate either the color or the fruit. Furthermore, it is challenging to distinguish "backpack" visually from other types of bags. To ensure the accuracy of the evaluation, we removed these terms from the lists.

We used prompts in Table A2 for Experiment (iii).

## B ADDITIONAL RESULTS

**Visualization of Attention Maps**   We visualized the attention maps linked to the words of interest in a given prompt in Fig. A1. The attention map is obtained by applying a softmax operation along the token (word) axis of the feature map of the CNN. Consequently, at most, one attention map can respond strongly at a specific pixel location. At the start of the reverse process ($t = 50$), each attention map responded to the image $x$ to some extent. As depicted in the left half of Fig. A1, the vanilla Stable Diffusion largely maintained the intensity distributions of the attention maps until the end of the reverse process, $t = 0$. This implies that the layout of the generated image is largely influenced by random initialization, which may not accurately capture the intended meaning of the prompt. The attention map for "bear" dominates the entire space at $t = 50$ and leaves no room for the attention map for "girl" to respond, resulting in the failure to generate a girl in the image. In contrast, given the proposition $\exists x.\ Girl(x)$, Predicated Diffusion encourages the attention map for "girl" to respond, even if it means dampening the response of the attention map for "bear", thereby ensuring the presence of a girl.

**Detailed Results of Experiment (iii)**   For reference, we summarize the actual numbers out of 20 responses for each prompt in the right half of Table A2. The successful rate varies greatly depending on the prompt, but the proposed method, Predicated Diffusion, shows the best results in almost all combinations of criteria and prompts. The sole exception is the prompt "A man holding a rabbit,"

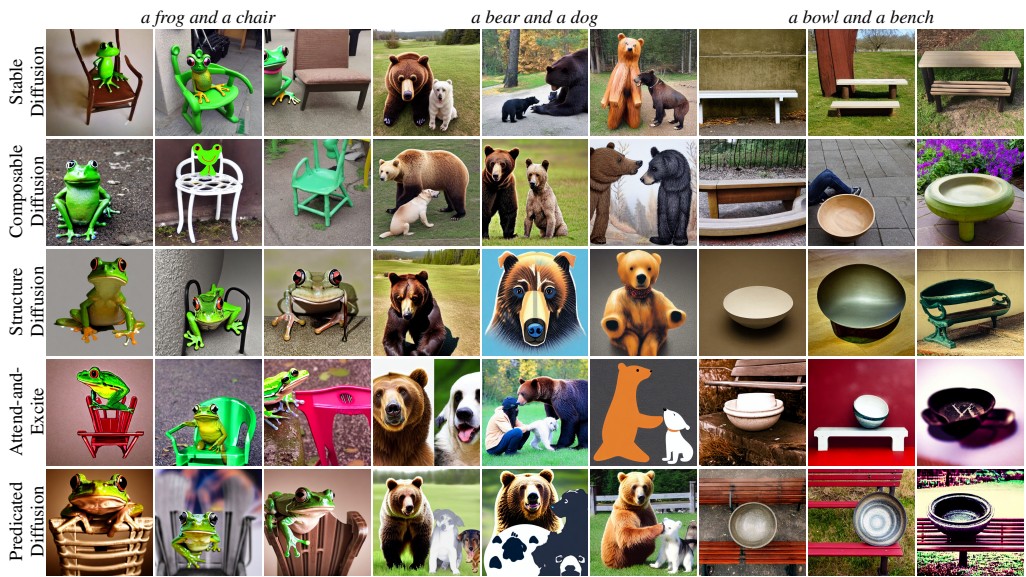

Figure A2: Additional results of Experiment (i) for concurrent existence. See also Fig. 3.

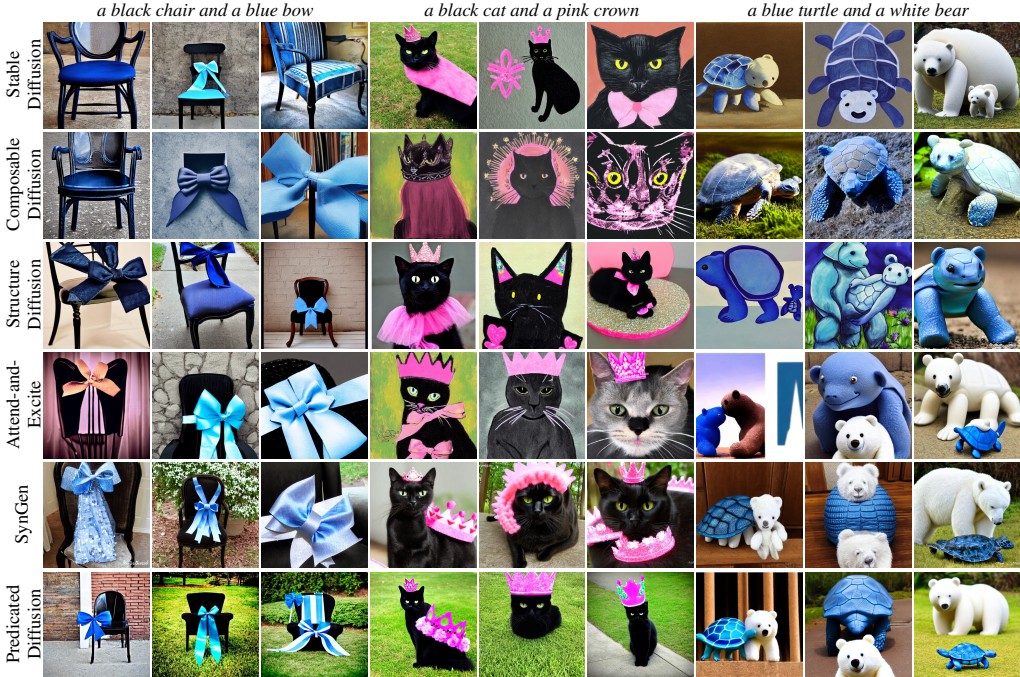

Figure A3: Additional results of Experiment (ii) for one-to-one correspondence. See also Fig. 4.

where Stable Diffusion already produced satisfactory results but Predicated Diffusion deteriorated the scores. When the backbone, Stable Diffusion, can generate images faithful to the prompt, the additional guidance might disturb the generation process.

**Additional Visualizations and Discussion**    We summarize the additional visual examples of Experiments (i)–(iv) in Figs. A2–A6. While a quantitative comparison is difficult, when Stable Diffusion produces satisfactory results, Predicated Diffusion often retains the original layout. This is because, as long as logical operations like implications are satisfied, Predicated Diffusion does not trigger any further changes.

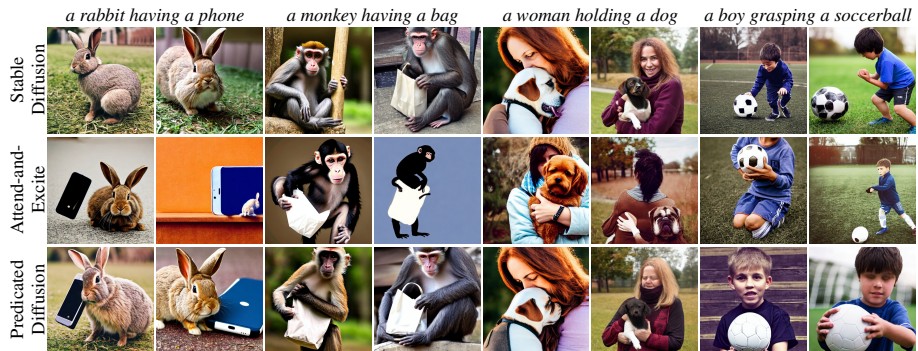

Figure A4: Additional results of Experiment (iii) for possession. See also Fig. 5 in the main body.

**Multi-color by Logical Disjunction** Consider cases where multiple colors are specified for a single object. For instance, the prompt "a green and grey bird" implies that every part of the bird is either green or grey, not both. This statement can be represented using the disjunction as $\forall x. Bird(x) \rightarrow Green(x) \vee Grey(x)$. The corresponding loss function is:

$$\mathcal{L}[\forall x. Bird(x) \rightarrow Green(x) \vee Grey(x)] = -\sum_i \log(1 - A_{Bird}[i] \times (1 - A_{Green}[i]) \times (1 - A_{Grey}[i])). \quad (A1)$$

When another object is introduced, one can replace the implication with a biimplication, as is the case with one-to-one correspondence.

The right six columns of Fig. A5 show example results. SynGen produced birds with a mixed hue because it was designed to equalize the intensity distributions (that is, the regions) of both specified colors and that of the bird. Conversely, Predicated Diffusion, based on predicate logic, can generate birds in the given combination of colors.

Note that, when multiple adjectives modify the same noun independently, they can be represented using the logical conjunction rather than the logical disjunction. For instance, the prompt "long, black hair" can be decomposed into two statements that can hold simultaneously: "The hair is long" and "The hair is black.". Then, the prompt is represented by the conjunction of two propositions that represent these statements.

**Negation by Logical Negation** In the right three columns of Fig. A6, we explored the negation of a concept. Sometimes, we might wish for certain concepts to be absent or negated in the generated images. If given a prompt "a polar bear", the output will typically be an image of a polar bear depicted with snowy landscapes because of their high co-occurrence rate in the dataset. One can give the prompt "a polar bear without snow", but Stable Diffusion often struggles to remove the snow, as depicted in the top row. Alternatively, we could provide a negative prompt "snow", as proposed as part of Composable Diffusion (Liu et al., 2022). We also examined Perp-Neg, which ensures a negative prompt not to interfere with a regular prompt by projecting the former's update to be orthogonal to the latter's update (Armandpour et al., 2023); it often failed to remove the snow. We consider an alternative way using predicate logic. The absence of snow is represented by the proposition $\neg(\exists x. Snow(x)) = \forall x. \neg Snow(x)$, leading the loss function

$$\mathcal{L}[\neg(\exists x. Snow(x))] = -\sum_i \log(1 - \bar{A}_{Snow}[i]), \quad (A2)$$

where $\bar{A}_c$ represents the attention map corresponding to a word $c$ in an auxiliary prompt like a negative prompt. This approach did not show any clear advantages compared to the negative prompt but at least demonstrated the generality of Predicated Diffusion.

**Ablation Study for Implication** In Section 3, we used implications to represent the modification by adjectives and the possession of objects as $\forall x. Noun(x) \rightarrow Adjective(x)$ and $\forall x. Object(x) \rightarrow Subject(x)$, respectively. In this section, we explored the effects of reversing the direction of these implications as an ablation study.

Figure A7 summarizes the results for modification. In the first row, a noun implicates an adjective, indicating that the object represented by the noun is uniformly colored by the hue represented by

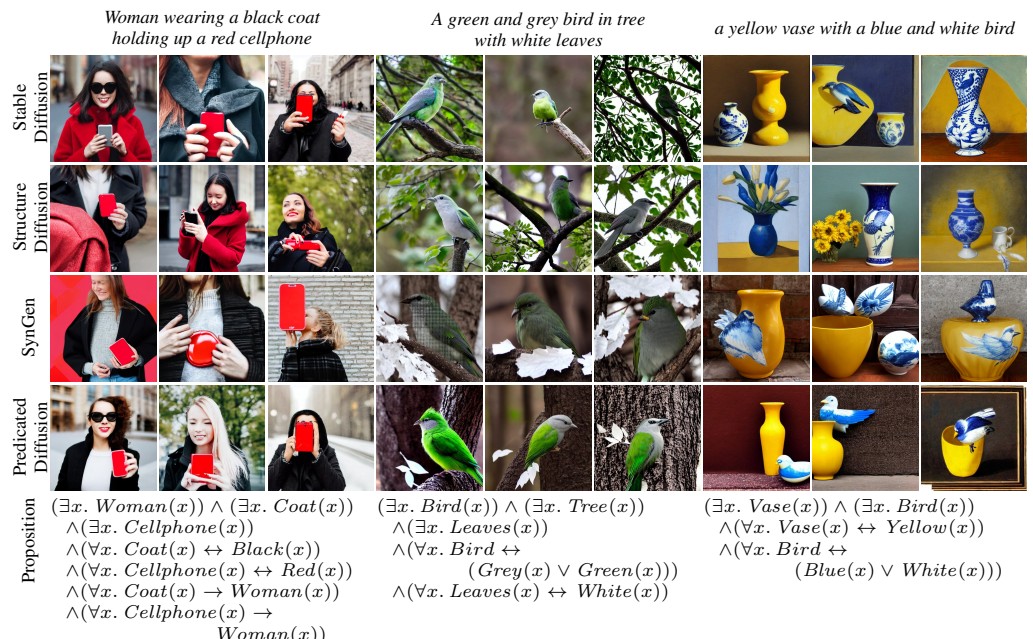

Figure A5: Example results of Experiment (iv) using prompts in ABC-6K. See also Figs. 6 and A6.

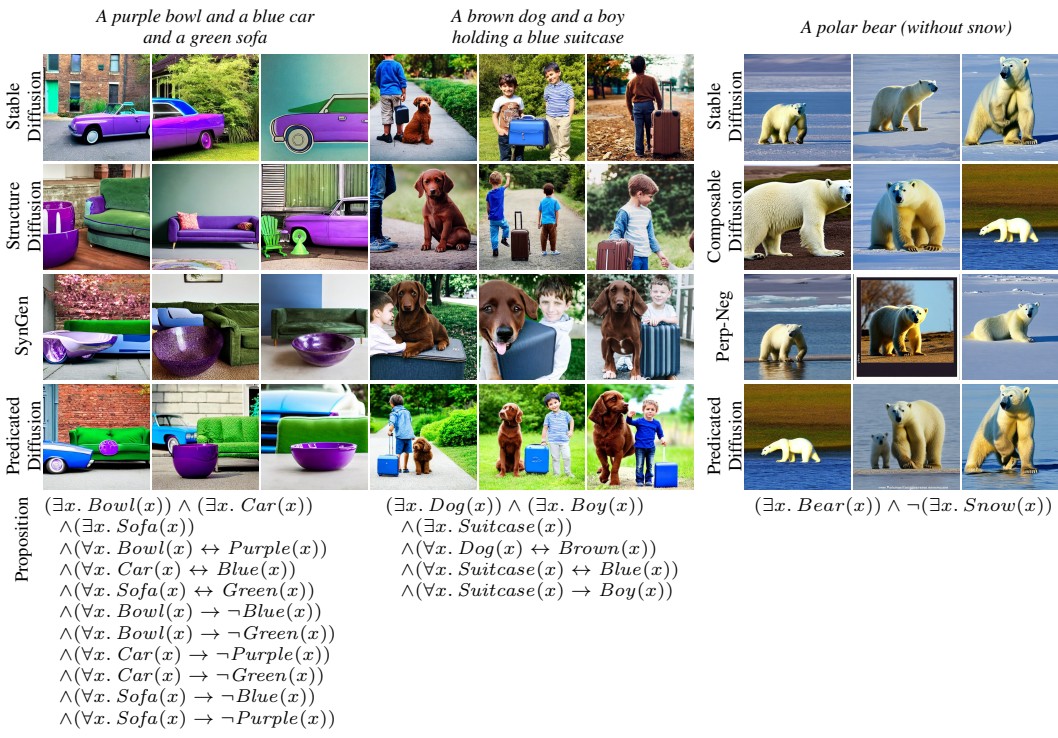

Figure A6: Example results of Experiment (iv) using our original prompts. See also Figs. 6 and A5.

the adjective. This allows other objects, such as the background, to share the same color. Conversely, when an adjective implicates a noun, the area colored by the adjective becomes a subset of that of the object, suggesting partial coloring of the object, as shown in the second row. With a biimplication, the color and object regions perfectly overlap. Therefore, biimplication is preferable to avoid attribute leakage, though implication may be useful depending on the context. A clear correspondence can be found between the semantics of propositions and the generated results.

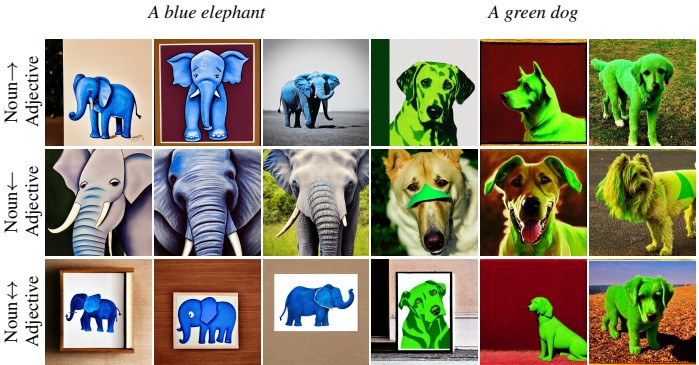

Figure A7: Ablation study for the adjective by the implication.

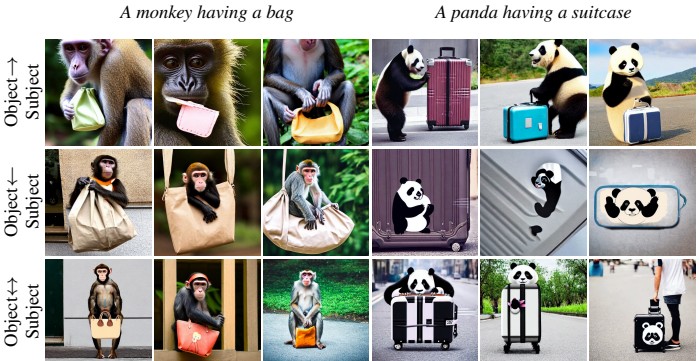

Figure A8: Ablation study for the possession by the implication.

Figure A8 summarizes the results for possession. In the first row, a grammatical object implicates a grammatical subject, and the results show the subject in possession of the object. Conversely, the second row shows that when the subject implicates the object, the subject becomes part of the object. For example, instead of "A monkey having a bag," the situation resembles "A bag envelops a monkey." Similarly, instead of "A panda having a suitcase," the scene is more like "A panda serves as a pattern on the suitcase." In cases of biimplication, the subject and object are often mixed together to form one object. Thus, implicating the subject by the object most accurately represents the subject in possession of the object.

## C   FOR AUTHOR RESPONSE



*A lion and a dog*        *A horse and a rabbit*

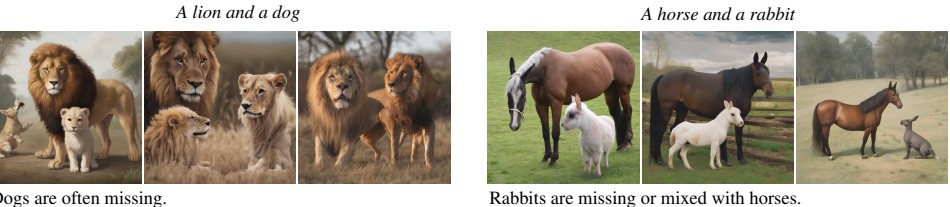

Dogs are often missing.        Rabbits are missing or mixed with horses.

*A blue frog and a green dog*        *A red bird and a yellow bowl*

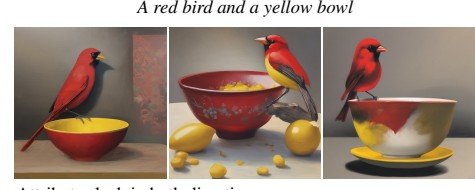
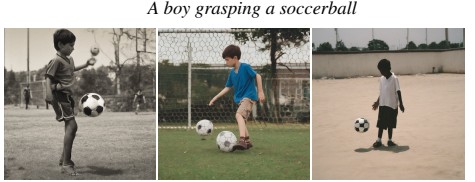

Dogs are missing or mixed with frogs. Frogs are sometimes        Attributes leak in both directions.
painted green.

*A boy grasping a soccerball*        *A panda having a suitcase*

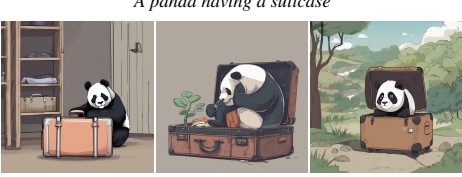

Soccer balls are never grasped.        Pandas tend not to have suitcases but to stay in them.



Figure A9: Example results of Stable Diffusion XL for author response.

