# OpenReview forum: "Predicated Diffusion: Predicate Logic-Based Attention Guidance for Text-to-Image Diffusion Models"
_ICLR.cc/2024/Conference — ICLR 2024 Conference Withdrawn Submission_

### Official Review · Reviewer_YsNP · 2023-10-31

**Soundness:** 3 good
**Presentation:** 2 fair
**Contribution:** 3 good
**Rating:** 5
**Confidence:** 4

**Summary:**

This paper proposed Predicated Diffusion, a comprehensive framework designed to articulate users' intentions effectively. This approach leverages predicate logic and utilizes pixels within attention maps as fuzzy predicates, with propositions serving as the textual representation. By employing this methodology, it transforms these intentions into a differentiable loss function. The experimental findings demonstrate a heightened faithfulness to the provided prompts. Furthermore, the paper introduces the concept of 'possession failure,' which expands the scope of inquiry to encompass the existence or non-existence of objects and attributes.

**Strengths:**

1.By using the Predicate Logic method, this paper easily converts the intentions of prompts into a differentiable loss function, which is a simple, intuitive, and effective method.

2.This paper proposes the term “possession failure” to describe the situation of missing attributes of prompts in T2I models. It provides a detailed direction for modeling the missing attributes question.

**Weaknesses:**

1. This paper represents an increment in the field, integrating additional predicate logics into the Attend-and-Excite framework. However, it's worth noting that the quality of the generated images appears to be subpar. For instance, Figure 1 illustrates issues such as a blurred bird in the first column, a cat with distorted textures in the second column, and a figure with missing eyes in the last column. These results suggest that the introduction of additional loss functions may have had a detrimental effect on the original model. Thus, a crucial question arises: How can we retain the benefits of the original model while addressing these shortcomings?

2. Despite the proposed method, the problem of "attribute leakage" persists. For example, in Figure 3, the model still generates two apples in response to the prompt "an apple and a lion," and this issue remains evident in Figure 4 with the prompt "a green balloon and a purple clock."

3. While this method effectively addresses the "possession failure" issue, it primarily focuses on Stable-Diffusion v1.4, rather than the latest SDXL model or the DALLE3 model. As a result, there may be limited instances of "possession failure," prompting a need to evaluate the overall contribution of this paper.

4. The paper conducts four distinct experiments to showcase the effects of integrating predicate logics. However, the question remains: If all predicate logics were learned simultaneously, what impact would this have on the original model's performance? Could it further deteriorate the model's results?

5. The proposed method lacks a discussion of its limitations. It is imperative to address and acknowledge the limitations of this approach in order to provide a comprehensive evaluation.

6. Several typographical errors require attention, such as the statement, "the existing models rarely fail to generate an object with a specified color." In subsection "One-to-One Correspondence" of the "METHOD" section, it appears that the intended message may be the opposite. Please clarify this statement for greater clarity.

**Questions:**

Please see the weaknesses.

---

> ### Author Response · Authors · 2023-11-13
> **We hope that our explanation will address your concerns (1/2)**
>
> Thank you for your comprehensive review and valuable suggestions. We are grateful for your recognition of the simplicity, intuitiveness, and effectiveness of our method. To address your concerns and questions, we have attached our responses below and will update the final manuscript accordingly.
>
> **1. This paper represents an increment in the field, integrating additional predicate logics into the Attend-and-Excite framework. However, it's worth noting that the quality of the generated images appears to be subpar. (...) How can we retain the benefits of the original model while addressing these shortcomings? (...)**
>
> **2. Despite the proposed method, the problem of "attribute leakage" persists. (...)**
>
> We understand your concerns regarding the generation quality. In our presentation of results, we consciously avoided cherry-picking only the most visually appealing images. Instead, we chose to showcase both the successes and limitations of our method. Our method does not completely resolve the issues but reduces their occurrences while keeping image quality.
>
> To objectively evaluate the generation quality, we have performed the CLIP image quality assessment (CLIP-IQA), proposed in (Wang et al., AAAI, 2023). We summarized the results in the following table.
>
> CLIP image quality assessment (higher is better)
>
> | Models               | Experiment (i) | Experiment (ii) | Experiment (iii) |
> |:---------------------|:---------------|:----------------|:-----------------|
> | Stable Diffusion     | 0.761          | 0.756           | 0.762            |
> | Composable Diffusion | 0.764          | 0.757           | --               |
> | Structure Diffusion  | 0.763          | 0.760           | --               |
> | Attend-and-Excite    | 0.766          | 0.761           | 0.760            |
> | SynGen               | --             | 0.750           | --               |
> | Predicated Diffusion | **0.775**      | **0.769**       | **0.765**        |
>
> This metric evaluates whether the image is closer to the text "good photo" or "bad photo" in the embedding space, and has been proven to be highly correlated with the human perception of image quality. It does not require the text prompts used for image generation and measures purely the image quality. Predicated Diffusion achieved the best scores in all three Experiments. Therefore, we can conclude that the proposed Predicated Diffusion rather improved the image quality, while SynGen degraded it.
>
> For the fidelity of generated images to the text prompts, we measured similarities between text prompts and generated images using CLIP and BLIP. The results, presented in Tables III and IV, demonstrate the superiority of our method.
>
> (Wang et al., AAAI, 2023) Jianyi Wang, Kelvin C.K. Chan, Chen Change Loy, Exploring CLIP for Assessing the Look and Feel of Images, AAAI Conference on Artificial Intelligence, 2023.
>
> **3. While this method effectively addresses the "possession failure" issue, it primarily focuses on Stable-Diffusion v1.4, rather than the latest SDXL model or the DALLE3 model.**
>
> The studies of these models are concurrent with ours, making integration challenging. For reference, we have included some images generated using Stable Diffusion XL at the end of the revised manuscript. These additional images demonstrate that Stable Diffusion XL still faces the same issues including the possession failure. Therefore, we believe that our method is helpful even for cutting-edge models.
>
> **4. (...) If all predicate logics were learned simultaneously, what impact would this have on the original model's performance?**
>
> Experiment (i), using Eq. (2), is a subset of Experiment (ii), using Eqs. (2) and (5), or Experiment (iii), using Eqs. (2) and (6). All three cases of Experiment (iv) in Fig. 6 simultaneously used Eqs. (2), (5), and (6) and succeeded in generating a variety of faithful and high-quality images.
>
> Note that, in Experiment (ii), we have already incorporated eight different propositions (as listed below). In Experiment (iv), sometimes more than ten propositions are incorporated, and no particular problems have been observed. Thus, our experiments suggest that even with the simultaneous learning of many propositions, the model's performance remains robust and effective.
>
> Eight different propositions: $\exists x. Dog(x)$, $\exists x. Cat(x)$, $\forall x. Dog(x)\rightarrow Black(x)$, $\forall x. Black(x)\rightarrow Dog(x)$, $\forall x. Cat(x)\rightarrow White(x)$, $\forall x. White(x)\rightarrow Cat(x)$, $\forall x. Dog(x)\rightarrow \neg White(x)$, and $\forall x. Cat(x)\rightarrow \neg Black(x)$

---

> ### Author Response · Authors · 2023-11-13
> **We hope that our explanation will address your concerns (2/2)**
>
> **5. The proposed method lacks a discussion of its limitations.**
>
> Throughout the manuscript, we showcased both the successes and failures of our method in figures, demonstrating that it does not completely resolve the issues but reduces their occurrences. As discussed on page 16, when the backbone, Stable Diffusion, can generate images faithful to the prompt, the additional guidance might disturb the generation process. Although this is not a frequent occurrence, as evidenced by the overall improvement in average CLIP-IQA scores, we acknowledge these instances. Furthermore, our method inherits some of the general limitations of Stable Diffusion. These include challenges like the inability to count objects accurately and a tendency for bias towards typical examples. These issues will be addressed in future work.
>
> **6. Several typographical errors require attention, such as the statement, "the existing models rarely fail to generate an object with a specified color." In subsection "One-to-One Correspondence" of the "METHOD" section, it appears that the intended message may be the opposite. Please clarify this statement for greater clarity.**
>
> Sorry for the confusion. What we intended to express was the following: 'If a text prompt specifies only one object and one color, then the vanilla Stable Diffusion works fine in most cases, and hence the loss function (3) is not needed. Therefore, this section focuses on cases where multiple objects and multiple colors are specified.'
>
> To eliminate unclear explanations and improve readability, we will restructure and revise the entire manuscript for the final draft.

---

### Official Review · Reviewer_5oLz · 2023-10-31

**Soundness:** 3 good
**Presentation:** 2 fair
**Contribution:** 3 good
**Rating:** 6
**Confidence:** 3

**Summary:**

The paper introduced a novel approach to guide text-to-image diffusion models in order to improve relation consistency within a generative image given a prompt. To this end, the introduced guidance links predicate logic and attention maps of diffusion models. The authors motivated the utilization of logic based on four issues, namely missing objects, unintended mixture of objects, attribute leakage between objects, and possession failure. After introducing the methodology, these issues are used to assess the performance of the proposed guidance and other baselines.

**Strengths:**

- The paper addresses a currently unresolved issue of text-to-image diffusion models.
- The issues addressed are well explained and motivated.
- The implementation of propositions via attention maps is well introduced and well supported with examples, which makes it easy to understand.
- While the evaluation only considers Stable Diffusion, the approach can be transferred to other diffusion models without the need to adapt parameters and any additional training.

**Weaknesses:**

- While the methodology is well introduced, the experiments lack clarity.
	- It is, for example, unclear how the prompts were selected. Are they extracted from existing datasets?
	- In the case of experiment 3, why is the similarity metric missing?
	- In Tables 3 and 4, is the fidelity corresponding to the rating from the user study? Are the reported values normalized? The authors describe that fidelity is assessed by human evaluators and two automated similarity approaches. However, it is not clear to which column these different metrics correspond. Further, it is unclear how the human ratings were aggregated; how many images were assessed by each human evaluator? What is reported, e.g., majority decision?

- The limitations are not well discussed. E.g., the compute overhead of the additional predicate logic-based guidance is unclear. Compared to, e.g., autoregressive image generative models, diffusion models’ inference time is rather slow. While approaches exist tackling these issues, I assume that the additional guidance introduced increases computation.

- Missing related work:
	- Universal Guidance for Diffusion Models. Arpit Bansal, Hong-Min Chu, Avi Schwarzschild, Soumyadip Sengupta, Micah Goldblum, Jonas Geiping, Tom Goldstein. CVPR Workshops 2023.
	- SEGA: Instructing Diffusion using Semantic Dimensions. Manuel Brack, Felix Friedrich, Dominik Hintersdorf, Lukas Struppek, Patrick Schramowski, Kristian Kersting. In Proceedings of NeurIPS 2023


Minor comment:

Typo: Section 3 second paragraph "Predicate Logic in Attention Map and Resulting Gauidance" -> Resulting Guidance

**Questions:**

Next to the questions raised above:

- Can you provide the computation costs you observed in your experiments, especially the additional overhead of using the introduced guidance?

- Which Stable Diffusion version is used in the experiments?

- You mentioned that the text encoder causes the addressed issues. Did you evaluate your method on diffusion models not relying on the CLIP text encoder and instead using, e.g., a more complex LM such as T5? For example, IF or Stable Diffusion XL? And could the introduced guidance be utilized during training or fine-tuning the text encoder?

- Why is the fidelity increasing when using Predicate Diffusion? Is this because of resolving the issues of object mixtures?
- How were the human ratings aggregated?
- Can you provide more details on the conducted user study? How many images were assessed by each human evaluator? What is reported, e.g., majority decision? Why are eight raters an appropriate and sufficient number of participants?

---

> ### Author Response · Authors · 2023-11-13
> **Thank you very much for your important suggestions (1/2)**
>
> Thank you very much for your detailed review and important suggestions. We are very pleased that you find our target issues well explained and motivated and the implementation well introduced and supported. To address your concerns and questions, we have attached our responses below and will update the final manuscript accordingly.
>
> **It is, for example, unclear how the prompts were selected. Are they extracted from existing datasets?**
>
> We explained that in Appendix A.2. In Experiments (i) and (ii), we followed the experimental settings used in the Attend-and-Excite paper, with minor modifications. Roughly speaking, we randomly selected nouns and adjectives from candidates for every generation. The SynGen paper also conducted similar experiments. In Experiments (iii), we manually selected prompts to ensure diversity.
>
> **In the case of experiment 3, why is the similarity metric missing?**
>
> We initially considered that automatic evaluations were not suitable for our self-prepared prompts. However, to improve clarity and objectivity in Experiment (iii), we will incorporate the following table that includes the similarity metrics and quality measures (CLIP-IQA). The results demonstrate that our method achieved better similarities and quality than comparison methods.
>
>
> | Models               | similarity            | quality (CLIP-IQA) |
> |:---------------------|:----------------------|:-------------------|
> | Stable Diffusion     | 0.320 / 0.811         | 0.762              |
> | Attend-and-Excite    | 0.334 / 0.843         | 0.760              |
> | Predicated Diffusion | **0.345** / **0.855** | **0.765**          |
>
> CLIP-IQA was proposed in Jianyi Wang, Kelvin C.K. Chan, Chen Change Loy, Exploring CLIP for Assessing the Look and Feel of Images, AAAI Conference on Artificial Intelligence, 2023.
>
>
> **In Tables 3 and 4, is the fidelity corresponding to the rating from the user study? (...) However, it is not clear to which column these different metrics correspond. (...) How were the human ratings aggregated?**
>
> We appreciate your suggestion for clarity. We will add appropriate headers to Tables 3 and 4 for readability.
>
> Each image was evaluated by a single human evaluator. This approach was chosen to prioritize evaluating a larger number of images rather than stabilizing the evaluation of a single image. We assigned 3, 3, and 2 human evaluators for Experiments (i), (ii), and (iii), respectively. Each evaluator assessed the same number of images; e.g., images generated using 133-134 prompts and five models in Experiment (ii).
>
> **The limitations are not well discussed. E.g., the compute overhead of the additional predicate logic-based guidance is unclear.**
>
> The computational overhead of attention guidance methods (Attend-and-Excite, SynGen, and ours) is approximately 25% of the main Stable Diffusion process. This holds regardless of the number of loss functions used. Typically, Stable Diffusion employs classifier-free guidance, needing the computation of the full U-Net with and without a prompt. Therefore, the total computational cost for Stable Diffusion is double that of U-Net. Since Stable Diffusion already generates attention maps during its reverse process, attention guidance methods can use these pre-existing maps to define loss functions, incurring negligible additional computational cost. To update the image, the gradient of the total loss function is backpropagated through the first half of U-Net, as the attention mechanism is located in the middle layer. This process incurs a computational cost about half that of U-Net. Given these considerations, we can estimate the additional computational overhead introduced by attention guidance methods to be around 25%.
>
> Furthermore, our method inherits some of the general limitations of Stable Diffusion. These include challenges like the inability to count objects accurately and a tendency for bias towards typical examples. These issues will be addressed in future work.

---

> > ### Author Response · Authors · 2023-11-13
> > **Thank you very much for your important suggestions (2/2)**
> >
> > **Missing related work**
> >
> > We acknowledge your recommendations and will cite the suggested papers in our related work section. The first paper uses additional classification and segmentation models for guidance, and the second one explores semantic directions in image generation. While these studies are not directly comparable to ours, their contributions warrant mention.
> >
> > **Which Stable Diffusion version is used in the experiments?**
> >
> > In our experiments, we used Stable Diffusion v1.4. This will be specified in our revised manuscript.
> >
> > **You mentioned that the text encoder causes the addressed issues. Did you evaluate your method on diffusion models not relying on the CLIP text encoder and instead using, e.g., a more complex LM such as T5? For example, IF or Stable Diffusion XL? And could the introduced guidance be utilized during training or fine-tuning the text encoder?**
> >
> > The studies of these models are concurrent with ours, making integration challenging. However, our method is viable as long as the backbone uses the attention mechanism. For reference, we have included some images generated using Stable Diffusion XL at the end of the revised manuscript. These additional images demonstrate that Stable Diffusion XL still faces the same issues we have examined. Therefore, we believe that our method is helpful even for cutting-edge models.
> >
> > We did not initially consider this, but our method potentially contributes to training or fine-tuning the text encoder as a regularization term. We will mention this as a possibility for future work.
> >
> > **Why is the fidelity increasing when using Predicate Diffusion? Is this because of resolving the issues of object mixtures?**
> >
> > Yes, partially. Fidelity in this context refers to how faithfully the generated images adhere to the text prompt. Therefore, successfully avoiding issues such as object mixture or missing objects directly contributes to improved fidelity.
> >
> > In the case of similarities (automatic evaluation of fidelity), CLIP and BLIP may struggle to extract appropriate embedding vectors from low-quality images. Consequently, the image quality potentially influences the similarities.
> >
> > To objectively evaluate the generation quality, we have performed the CLIP image quality assessment (CLIP-IQA), proposed in (Wang et al., AAAI, 2023). We summarized the results in the following table.
> >
> > CLIP image quality assessment (higher is better)
> >
> > | Models               | Experiment (i) | Experiment (ii) | Experiment (iii) |
> > |:---------------------|:---------------|:----------------|:-----------------|
> > | Stable Diffusion     | 0.761          | 0.756           | 0.762            |
> > | Composable Diffusion | 0.764          | 0.757           | --               |
> > | Structure Diffusion  | 0.763          | 0.760           | --               |
> > | Attend-and-Excite    | 0.766          | 0.761           | 0.760            |
> > | SynGen               | --             | 0.750           | --               |
> > | Predicated Diffusion | **0.775**      | **0.769**       | **0.765**        |
> >
> > This metric evaluates whether the image is closer to the text "good photo" or "bad photo" in the embedding space, and has been proven to be highly correlated with the human perception of image quality. It does not require the text prompts used for image generation and measures purely the image quality. Predicated Diffusion achieved the best scores in all three Experiments. Therefore, we can conclude that the proposed Predicated Diffusion improved the image quality, too.
> >
> > (Wang et al., AAAI, 2023) Jianyi Wang, Kelvin C.K. Chan, Chen Change Loy, Exploring CLIP for Assessing the Look and Feel of Images, AAAI Conference on Artificial Intelligence, 2023.
> >
> > **Typo**
> >
> > Sorry. We will address this and ensure our manuscript is proofread by experts.

---

### Official Review · Reviewer_2NoA · 2023-10-31

**Soundness:** 2 fair
**Presentation:** 3 good
**Contribution:** 3 good
**Rating:** 3
**Confidence:** 3

**Summary:**

This paper studies the misalignment between image and text in text-to-image generation. The paper proposes a framework that represents the input text prompt using predicate logic. The attention weight of each pixel is then considered as a continuous value that indicates the level of fulfillment of a pixel for a specific proposition. The intermediate image at each denoising step is then updated in order to maximize the level of fulfillment of the input prompt. Experiments show that the proposed method outperforms several baselines on generating more complete objects and objects with correct colors.

**Strengths:**

1. The paper proposes a novel framework for generating images that are faithful to the input text prompt. The framework is generic in that it covers various issues that have been studied in previous works, such as missing objects and mistakenly bonded colors.
2. The experiments show that the proposed method outperforms existing baselines on four evaluated settings.

**Weaknesses:**

1. Some of the assumptions that are used for representing text prompt as predicate logic do not make sense. For example, the prompt "There is a black dog" is interpreted as "There is a dog" AND "All dogs are black," which won't work for prompts such as "A black dog and a white dog." Similarly, prompts that have possession relationships such as "a man holding a bag" is interpreted as "all pixels of the bag is also part of the man," which is not necessarily correct.
2. The proposed optimization method will not guarantee that all predicates are satisfied. When multiple predicates exist in the text prompt, their conjunction is used as the objective function. However, since this is a multi-objective optimization problem, the optimization used in the paper is not guaranteed to find optimal solution for all predicates.
3. The visualized images in the paper seem to not have as good quality as the baselines. No metrics (either automatic ones such as FID or subjective evaluations) are reported in the paper.

**Questions:**

1. Should $P(x) \rightarrow Q(x)$ be $1-A_P[i] \times (1-A_P[i] \times A_Q[i])$?

---

> ### Author Response · Authors · 2023-11-13
> **We hope that our explanation will help you understand (1/2)**
>
> Thank you very much for your detailed review and important suggestions. We are very pleased that you find our method novel and generic. To address your concerns and questions, we have attached our responses below and will update the final manuscript accordingly.
>
> **1. Some of the assumptions that are used for representing text prompt as predicate logic do not make sense. For example, the prompt "There is a black dog" is interpreted as "There is a dog" AND "All dogs are black," which won't work for prompts such as "A black dog and a white dog."**
>
> The prompt "There is a black dog" can be interpreted as "There is a dog" AND "All dogs are black," only if it is the entire prompt.
> If your prompt includes other statements, you may require additional considerations. For the prompt "A black dog and a white dog," you can apply the loss function for one-to-one correspondence using the proposition $(\forall x. Dog_1(x)\leftrightarrow Black(x))\land (\forall x. Dog_2(x)\leftrightarrow White(x))$. Here, we represent the first and second occurrences of the word "dog" as $Dog_1$​ and $Dog_2$​, respectively. However, due to the limitations of the attention mechanism, the attention maps for $Dog_1$ and $Dog_2$​ may overlap, causing a failure to differentiate between the two dogs.
> This issue is not an intrinsic flaw of our method but reflects a broader challenge: diffusion models still struggle to accurately capture the number of instances, and addressing this remains an unresolved question in the field.
>
> **1. Similarly, prompts that have possession relationships such as "a man holding a bag" is interpreted as "all pixels of the bag is also part of the man," which is not necessarily correct.**
>
> We would like to clarify that our method does not strictly enforce "all pixels of the bag is also part of the man." We employed fuzzy logic, specifically product fuzzy logic, to capture tendencies rather than absolute properties. The attention maps are typically lower in resolution compared to the original image, which allows slight displacements in object positioning. This flexibility is evident in examples such as "a frog wearing a hat" or "a girl holding a suitcase" shown in Fig. A4. In these cases, the objects are close to but do not entirely overlap with their possessors. Additionally, our results in Fig. A8 demonstrate that the operator $\rightarrow$ more accurately represents the possession relationship compared to $\leftarrow$ or $\land$, further supporting the validity of our method.
>
> **2. The proposed optimization method will not guarantee that all predicates are satisfied.**
>
> Indeed, it is true, but not a problem. The reverse process of the diffusion model is similar to gradient descent in finite steps, which may converge to a local minimum or stop early. However, a local minimum is often sufficient to improve the fidelity and quality of the generated images. The challenge arises with poor initial values, which can lead to poor local minima. To mitigate this, we have introduced an iterative refinement at the beginning of the reverse process, as detailed in Appendix A.1. This process helps adjust the initial values for better outcomes. While we acknowledge that more advanced optimization methods are helpful, the current approach does not limit the performance of our method.

---

> ### Author Response · Authors · 2023-11-13
> **We hope that our explanation will help you understand (2/2)**
>
> **3. The visualized images in the paper seem to not have as good quality as the baselines. No metrics (either automatic ones such as FID or subjective evaluations) are reported in the paper.**
>
> Thank you for your valuable suggestion. Fréchet Inception Distance (FID) requires a dataset of real images, so it is unavailable for our Experiments, which do not have corresponding ground truth images. Instead, to objectively evaluate the generation quality, we have performed the CLIP image quality assessment (CLIP-IQA), proposed in (Wang et al., AAAI, 2023). We summarized the results in the following table.
>
> CLIP image quality assessment (higher is better)
>
> | Models               | Experiment (i) | Experiment (ii) | Experiment (iii) |
> |:---------------------|:---------------|:----------------|:-----------------|
> | Stable Diffusion     | 0.761          | 0.756           | 0.762            |
> | Composable Diffusion | 0.764          | 0.757           | --               |
> | Structure Diffusion  | 0.763          | 0.760           | --               |
> | Attend-and-Excite    | 0.766          | 0.761           | 0.760            |
> | SynGen               | --             | 0.750           | --               |
> | Predicated Diffusion | **0.775**      | **0.769**       | **0.765**        |
>
>
> This metric evaluates whether the image is closer to the text "good photo" or "bad photo" in the embedding space, and has been proven to be highly correlated with the human perception of image quality. It does not require the text prompts used for image generation and measures purely the image quality. Predicated Diffusion achieved the best scores in all three Experiments. Therefore, we can conclude that the proposed Predicated Diffusion rather improved the image quality, while SynGen degraded it.
>
> For the fidelity of generated images to the text prompts, we measured similarities between text prompts and generated images using CLIP and BLIP. The results, presented in Tables III and IV, demonstrate the superiority of our method.
>
> (Wang et al., AAAI, 2023) Jianyi Wang, Kelvin C.K. Chan, Chen Change Loy, Exploring CLIP for Assessing the Look and Feel of Images, AAAI Conference on Artificial Intelligence, 2023.
>
> **Q1. Should $P(x)\rightarrow Q(x)$ be $1-A_P[i]\times (1-A_P[i]\times A_Q[i])$?**
>
> Your expression aligns with classical Boolean logic but not product fuzzy logic. Yours can derive from the logical expression $\neg(P(x)\land(\neg(P(x)\land Q(x))))$, whereas ours derived from $\neg(P(x)\land \neg Q(x))$. These two expressions are equivalent in classical Boolean logic, but they yield different outcomes under product fuzzy logic. Our expression is based on the strong conjugation and material implication of the product fuzzy logic, as mentioned on page 4, and consistent with the theory underlying product fuzzy logic.

---

### Official Review · Reviewer_h1q7 · 2023-11-01

**Soundness:** 3 good
**Presentation:** 2 fair
**Contribution:** 3 good
**Rating:** 6
**Confidence:** 4

**Summary:**

The paper introduces Predicated Diffusion which combines predicate logic with the intuition of cross-attention layers in diffusion-based text-to-image. The paper draws connections between several propositions and attention map operations. Language prompts can be seen as a combination of these propositions and have corresponding loss functions that can be optimized in the diffusion process.  Experimental results show that the method outperforms several baselines including a recent SOTA method in this direction.

**Strengths:**

- The proposed method is novel. It is very interesting to see how first-order logic can be connected to compositionality in text-to-image generation, specifically the attention maps. Some of the propositions and losses are reasonable and interpretable.
- The proposed method tackles a wide range of problems, including well-studied ones and also an underaddressed problem, i.e. possession failure.
- Experimental results show that the method outperforms previous methods in many aspects.

**Weaknesses:**

- Some of the losses are not intuitive or cannot be easily verified. I am not sure if this is due to the presentation of the method section. For example, how does eq (2) prevent the two objects from highlighting the same pixels or regions? It would be better to give straightforward intuition behind the equations in terms of the behavior of attention maps. For example, if I understand correctly, eq 6 encourages the attention maps of "bag" to be partially overlapped with the attention maps of "man" yet does not force all pixels of "bag" to be part of the "man".
- Predicated Diffusion requires manual or pre-defined use of different propositions for different prompts. As stated in Sec. 4, the authors applied different losses for different types of prompts. However, this is not practical for applications where prompts can be arbitrary. The authors manually extracted propositions for each prompt in Experiment (iv) which, I think, really downgrades the overall value of the work. Is there an automatic way to extract propositions for each prompt?
- Writing could be improved. I think Sec. 3 could be improved in structure and contents. Some paragraphs have too many logical equations that make them a bit hard to follow. Perhaps the authors could find a more organized way to explain every proposition (e.g. start with a simple derivation of logic equations, then provide the attention equation, and finally give some intuition in words. ). The authors could attempt to group contents into subsections to illustrate propositions from easy ones to hard ones and distinguish the novel propositions over A&E or SynGen. There are other trivial flaws like using "Experiment (x)" in tables/captions without specifying the experiment domain, making it hard to follow.

While I really like the novelty and perspectives presented by the work, there are major weaknesses. I will adjust my rating accordingly depending on how well these concerns are resolved.

**Questions:**

See above.

---

> ### Author Response · Authors · 2023-11-13
> **Thank you very much for your valuable suggestions.**
>
> Thank you very much for your detailed review and valuable suggestions. We are very pleased that you recognize the novelty and perspectives. To address your concerns and questions, we have attached our responses below and will update the final manuscript accordingly.
>
> **Some of the losses are not intuitive or cannot be easily verified.**
>
> Sorry for our unclear explanation. Please allow me to respond as follows.
>
> > **how does eq (2) prevent the two objects from highlighting the same pixels or regions?**
>
> The attention maps are generated through softmax operations, which inherently prevent the two objects from highlighting the same pixels or regions. When one object's attention map is activated, it naturally suppresses overlapping regions in other objects' maps. We will clarify this in the implementation details of the revised manuscript.
>
> > **eq 6 encourages the attention maps of "bag" to be partially overlapped with the attention maps of "man" yet does not force all pixels of "bag" to be part of the "man".**
>
> Eq. (6) DOES force all pixels of "bag" to be part of the "man".
> If there exists a pixel of "bag" ($Bag(x)=True$), our method enforces the same pixel represents "man" ($Man(x)=True$) or suppresses the response to the word "bag" ($Bag(x)=False$).
> If a pixel does not respond to "bag" ($Bag(x)=False$), our method does nothing.
> Please refer to the following True-False table:
>
> | Bag | Man | Bag->Man |
> |:----|:----|:---------|
> | F   | F   | T        |
> | F   | T   | T        |
> | T   | F   | F        |
> | T   | T   | T        |
>
> Due to the softmax operations mentioned above, the pixel intensities in both attention maps never reach 1.0 but instead converge to moderate values, as shown in Fig. A1. In practice, Eq. (6) encourages overlap but does not completely enforce it. Our goal is to guide the image generation process to fulfill the intended meaning; this implementation is deemed sufficient for that purpose.
>
> **Predicated Diffusion requires manual or pre-defined use of different propositions for different prompts. (...) Is there an automatic way to extract propositions for each prompt?**
>
> Yes, one can employ a syntactic dependency parser to automatically identify key words. For validation, we used spaCy v3.0 to extract:
>
> - Nouns for propositions of concurrent existence in Eq. (2),
> - Modifier-noun pairs for propositions of one-to-one correspondence in Eq. (5), and
> - Subject-object pairs connected by verbs indicating possession (have, hold, grasp, and wear, in this experiment) for propositions of possession in Eq. (6).
>
> For example, from the prompt "Woman wearing a black coat holding up a red cellphone," we extracted:
>
> - Nouns: Woman, coat, cellphone
> - Modifier-noun pairs: [black, coat], [red, cellphone]
> - Subject-object pairs: [Woman, coat], [Woman, cellphone]
>
> These results enabled us to automatically create propositions using a simple script.
> In Figs. 6 and A5, we successfully extracted all propositions (shown below the images) from the prompts, with two exceptions: the possession relationships in prompts "A black bird with a red beak" and "A white teddy bear with a green shirt and a smiling girl." In these cases, the preposition "with" expresses possession, but it can express merely existence in other cases. Hence, some prompts need more advanced syntactic analysis. Nonetheless, our findings generally support the sufficiency of simple syntactic analysis.
>
> Furthermore, we believe it is crucial for users to explicitly use predicate logic to clarify their intentions that cannot be fully expressed in text. The meaning of "with," whether indicating a possession relationship or merely concurrent existence, can sometimes be ambiguous even for human readers. By employing our proposed method, users can clearly express their intentions and eliminate such ambiguities.
>
> **Writing could be improved.**
>
> Thank you for your valuable suggestion. To improve the manuscript, we plan to restructure Section 3. We will group the content into subsections and provide supportive descriptions for tables and captions. Due to page limitations, some content will be moved to the Appendix.